# Modifying Phosphate Toxicity in Chronic Kidney Disease

**DOI:** 10.3390/toxins11090522

**Published:** 2019-09-09

**Authors:** Marc Vervloet

**Affiliations:** Department of Nephrology and Amsterdam Cardiovascular Sciences, Amsterdam University Medical Center, De Boelelaan 1117, 1081 HV Amsterdam, The Netherlands; m.vervloet@amsterdamumc.nl; Tel.: +31-20-444-2671

**Keywords:** chronic kidney disease, CKD-MBD, phosphate

## Abstract

Phosphate toxicity is a well-established phenomenon, especially in chronic kidney disease (CKD), where hyperphosphatemia is a frequent occurrence when CKD is advanced. Many therapeutic efforts are targeted at phosphate, and comprise dietary intervention, modifying dialysis schemes, treating uncontrolled hyperparathyroidism and importantly, phosphate binder therapy. Despite all these interventions, hyperphosphatemia persists in many, and its pathological influence is ongoing. In nephrological care, a somewhat neglected aspect of treatment—when attempts fail to lower exposure to a toxin like phosphate—is to explore the possibility of “anti-dotes”. Indeed, quite a long list of factors modify, or are mediators of phosphate toxicity. Addressing these, especially when phosphate itself cannot be sufficiently controlled, may provide additional protection. In this narrative overview, several factors are discussed that may qualify as either such a modifier or mediator, that can be influenced by other means than simply lowering phosphate exposure. A wider scope when targeting phosphate-induced comorbidity in CKD, in particular cardiovascular disease, may alleviate the burden of disease that is the consequence of this potentially toxic mineral in CKD.

## 1. Introduction

For a long time, phosphate, especially when its concentrations are relatively high, is recognized as a potential toxic substance for human health [1]. Since the kidney is the principle organ regulating phosphate homeostasis, deranged serum phosphate concentrations are encountered especially in the setting of kidney diseases. Several monogenetic diseases, that impair tubular reabsorption of ultrafiltered phosphate, induce hypophosphatemia due to renal losses [2]. These conditions are relatively rare and the key clinical features are rickets and osteomalacia. In turn, phosphate retention may occur with loss of glomerular filtration rate (GFR). In this setting, tubular phosphate reabsorption is blocked, to ensure phosphate clearance. This inhibition of re-entering of phosphate into the circulation from pro-urine is accomplished by the compensatory increased concentration of two hormones, the parathyroid hormone (PTH), and fibroblast growth factor 23 (FGF23). However, as chronic kidney disease (CKD) progresses, this compensation falls short, and hyperphosphatemia develops. This metabolic consequence of CKD is considered to explain part of the increased mortality risk, with which CKD is associated [1,3,4]. Based on these observations, targeting hyperphosphatemia is considered a legitimate goal of treatment in CKD and guidelines do support such an approach [5]. Many therapeutic maneuvers in every day clinical practice are being instituted for this purpose, as exemplified by the observation that some fifty percent of total pill burden of patients on hemodialysis consists of phosphate lowering drugs [6]. Modes to control hyperphosphatemia or phosphate intake, including dietary intervention, phosphate binders, phosphate transport inhibitors, controlling hyperparathyroidism, restricted use of (high dosed) active vitamin D, and adapting dialysis schemes if applicable, have been reviewed recently and are beyond the scope of this review [7]. Despite these efforts to control serum phosphate concentrations, many patients on dialysis still do not reach treatment targets [8], and hence a treatment gap exists. This gap is an acknowledged unmet clinical need, which is addressed by increasing medication doses, adapting phosphate binder types, using combination therapy, intensifying dialysis schemes, and reevaluation by dietitians of intake of phosphate from diet and its specific sources [9]. In addition, since non-adherence is an important cause of treatment failure [10,11], it is suggested to improve self-management on an individual approach, but data are lacking that supports its effectiveness [12]. To make this matter even more frustrating, definite proof of the assumption that correcting hyperphosphatemia actually improves outcome is lacking, despite all these efforts, risk for side effects, burden of treatment and costs [5].

One possible way out of this stalemate situation might be to consider a different approach to mitigate the risk of a (uremic) toxin like phosphate. Since in toxicology, it is common practice to use an antidote when available, especially when it is not possible to sufficiently eliminate exposure, it is somewhat surprising that this treatment principle is, to a large extent, neglected when considering phosphate toxicity in CKD. To establish what might be the “antidote” in a setting of hyperphosphatemia, detailed consideration of how exactly phosphate inflicts harm, can provide relevant clues. Insight into factors that either mediate or modify phosphate toxicity, may be helpful in finding other ways than just targeting phosphate exposure, to alleviate morbidity induced by it. In this narrative review, several of these factors are highlighted, which might either be mediators or modifiers of phosphate toxicity as schematically depicted in Figure 1.

## 2. Formation of Calciprotein Particles

The physiological concentrations of phosphate and calcium in human plasma exceed its saturation product and hence an intrinsic propensity of these minerals to form crystals exists. Under physiological conditions, these crystals are covered by a protein monolayer consisting of Fetuin A together forming primary calciprotein particles (CPP1, Figure 2) [13]. In addition, small mineral clusters, also stabilized by Fetuin A, exist [13]. As such, these microcrystals are “scavenged” by Fetuin A, which presumably is a protective feature, because it prevents the formation of ectopic calcification on site where the microcrystals otherwise could have precipitated in growing hydroxyapatite structures, such as the cardiovascular system. However, in a pathological condition such as CKD, this system of scavenging calcium-phosphate nanocrystals can get overwhelmed, by either higher amounts of the minerals to be safety secured into the primary CPPs, by a reduction of the concentrations of fetuin A, or both. As fetuin A, produced by the liver, is a negative acute phase reactant, any condition of chronic inflammation, which also includes CKD, can mitigate the ability to pack the nanocrystals, because its circulating concentrations decline [14]. In this setting, free nanocrystals from calcium and phosphate may form, which, upon uptake my macrophages for instance resident in the vessel wall, induce local inflammation and drive formation of arterial calcification, much more so than when covered by fetuin-A [15]. Another typical feature of an overwhelmed CPP defense is the formation of secondary CPP (CPP2, Figure 2). This CPP2 has an entirely different shape than CPP1, probably because of formation of more mature and structured hydroxyapatite crystal formation, as a consequence of relative fetuin-A deficiency. In vitro studies have shown that CPP2 induce calcification of vascular smooth muscle cells (VSMC) [16]. Clinically, CPP2 is associated with arterial stiffness in predialysis CKD [17]. The concept of a disbalance between nanocrystal stress and its fetuin-A protection is supported by the early finding that low fetuin-A concentrations are associated with cardiovascular mortality in patients on dialysis [18]. More recently, in patients with CKD, the ratio of free fetuin-A over fetuin-A bound in serum mineral complex, was negatively associated with coronary artery calcification score, which also suggests that consumption of fetuin-A by nanocrystal stress, may induce vascular calcification [19].

Different from the direct measurement of the amount of CPP in blood samples or the total, or free fetuin-A concentration, is the measurement of the time that elapses to the formation of CPP2 from CPP1. This method has been developed and suggested to reflect the inherent protection against formation of the more toxic CPP2, and has been termed the T50, and is reflecting the time required for a given blood sample to have reached fifty percent of increased light scatter by nephelometry, indicating the formation of CPP2 [21]. Several observational studies have shown that this T50 score predicted all-cause mortality in both patients on dialysis and after kidney transplantation, importantly also so after adjustment for serum phosphate concentrations [22,23,24]. The intriguing hypothesis that emerges from these observations is that modifying T50, and thereby the formation rate of CPP2, improves nanocrystal stress, lowers vascular inflammation and calcification and ultimately clinical events. Factors that are of influence on the T50 are phosphate itself, low pH, calcium and fetuin-A deficiency (all shortening T50) and albumin, magnesium, fetuin-A, pyrophosphate and higher pH (all prolonging T50, assumedly beneficial) [25].

All these findings on the role of CPP’s, as physiological carriers of phosphate-calcium containing minerals, and the existence of an inherent protection against the formation of CPP2, gave rise to the hypothesis that not phosphate as such, but the nature of formed CPP, in particular CPP2 carry the risk of “phosphate toxicity” [20,26]. In that scenario, CPP formation, in particular CPP2, should be the target of treatment, and not serum phosphate concentration, because then phosphate toxicity would be mediated by the formation of CPP2.

## 3. FGF23 as Mediator of Phosphate Toxicity

The central physiological role of FGF23 is to maintain phosphate balance, by promoting its excretion by the kidneys, preventing activation of vitamin D, which limits phosphate uptake from the intestinal tract, and to inhibit PTH, which limits bone resorption and skeletal phosphate release [27]. By unresolved mechanisms, chronic exposure to phosphate, probably in parallel with calcium, stimulated FGF23 release from bone [28,29,30]. Therefore, it is, at least theoretically, possible that part of phosphate toxicity in fact is induced by a parallel increase of FGF23, especially in CKD where FGF23 increments are most pronounced [31]. Experimental data have shown that FGF23 can have a direct effect on the heart, by inducing left ventricular hypertrophy (LVH) [32,33], and pathologically interfering with calcium fluxes in cardiomyocytes out of and into the sarcoplasmic reticulum [34,35], both effects not depending on phosphate. This would be in line with the concept that FGF23 may be a mediator of phosphate toxicity, at least on the heart in a setting of CKD [36], and in that scenario targeting FGF23 by any means, besides addressing hyperphosphatemia, would be beneficial. This concept failed in an animal model of CKD where FGF23 blocking antibodies had increased arterial calcification and higher mortality compared to non-treated CKD animals [37]. This detrimental effect was ascribed to worsening hyperphosphatemia. Although currently burosumab, an antibody blocking FGF23 effects, is clinically available for patients with primary increased FGF23 associated with hypophosphatemia, [38,39], its use may be risky in CKD, because of concomitant aggravation of hyperphosphatemia. Therefore, other approaches to lower FGF23, without increasing serum phosphate concentrations, may seem more logical. In a post-hoc analysis of the EVOLVE trial (a placebo-controlled randomized trial among patients on hemodialysis, studying the effect of cinacalcet, an oral calcimimetic, on a composite of clinical endpoints) it was found that a cinacalcet-induced decline of FGF23 was associated with improved outcome [40]. There was a correlation coefficient of 0.519 between percentage decline of FGF23 and percentage decline of serum phosphate concentrations, and it remained difficult to exclude that to a large extent, any beneficial effect was not driven by phosphate reduction itself, because phosphate itself has also been shown to be able to induce myocardial hypertrophy [41].

Different from the development of left ventricular hypertrophy, where it is complicated in a clinical setting to disentangle effects of FGF23 and phosphate, the development of vascular calcification seems to be a more straightforward consequence of high phosphate exposure, but not of FGF23 [42,43]. Some studies do suggest that FGF23 itself, increased due to phosphate retention can pathologically change vasodilating capacity, possibly mediated by upregulation of asymmetrical dimethyl arginine (ADMA) [44,45], but the clinical relevance of this is uncertain, and phosphate itself has well established effects on the endothelium, independent of FGF23 [46,47,48].

Collectively, there are only limited data that suggest that the potency of phosphate to act as a uremic toxin, is very dependent on upregulation of FGF23. This implies that even if, in the future, it might be possible to selectively block untoward effects of FGF23, for instance on the heart [49], this will not dismiss the necessity to also target phosphate exposure. In turn, it might be possible that achieving phosphate control and simultaneously lower FGF23 can have synergistic effects, in particular on the heart.

## 4. Mitigating Effects of Magnesium on Phosphate Toxicity

Several large observational studies in patients on dialysis have consistently shown an association between higher serum magnesium concentrations in patients on dialysis and lower all-cause mortality, in particular, cardiovascular mortality [50,51]. This apparent protective effect of magnesium persisted after multivariable adjustments and was also shown in other populations, as recently summarized [52]. A most striking epidemiological finding was that the well-established U-shaped curve that describes the association between serum phosphate concentrations and cardiovascular mortality in patients on dialysis was very dependent on concurrent magnesium concentration [53], as shown in Figure 3. These data revealed that in a setting of higher magnesium concentration, the association of hyperphosphatemia and mortality was lost. The question arises what this striking effect of magnesium implies. It is possible that residual confounding may play an important role, as magnesium content in “healthy food” is much higher than in less healthy food, and that the improved survival is the consequence of a healthier lifestyle. Indeed, when adjusting for albumin concentration, as a proxy for nutritional status, the association between magnesium and mortality in patients on dialysis was lost [54]. In turn, it could be argued that correcting the presumed benefits of higher magnesium concentration for adequate nutritional status, might be overcorrection, because one of the benefits of a healthy diet might actually be its magnesium content. Another argument in favor of a direct role of magnesium itself are findings from trials with magnesium supplements, generally demonstrating protective effects indeed [55,56], as will be discussed below. Besides deciphering the potential modifying role of magnesium on hyperphosphatemia from epidemiological data, another approach could be to study if magnesium interferes with pathological changes on the tissue or cellular levels, induced by higher concentrations. Indeed, this provides additional arguments that magnesium may mitigate the procalcifying effects of higher phosphate concentrations. While high phosphate concentrations have been shown to induce a phenotypic switch of VSMC with a contractile phenotype into a secretory phenotype with features of bone forming cells [57,58], magnesium in turn prevents this from happening, thereby protecting form phosphate induced vascular calcification [59,60]. Besides these effects on the biological aspects of vascular calcification, magnesium also inhibits nanocrystal formation and growth from calcium and phosphate ions, a key aspect on the formation of ectopic calcification in CKD [61,62]. Interestingly, magnesium was also shown to act as an inhibitor of the formation rate (T50) of secondary CPPs [21], and clinically supplementation of magnesium in patients with different stages of CKD, both orally or from the dialysate by increasing its magnesium concentration, prolongs the T50-score [63,64]. A final interesting aspect of increased dietary intake of magnesium, is that it may chelate dietary phosphate and thereby act as a phosphate binder. Indeed, animal studies have shown this actually is the case and moreover attenuates the severity of vascular calcification in a CKD animal model [65]. Of note, in this study, supplementing magnesium via peritoneal injection also had some protective effects, indicating that gastrointestinal binding of phosphate was not its only mode of action.

Based on the above, magnesium appears to be an important modifier of phosphate-induced toxicity. Clearly, clinical prospective studies are needed to test the hypothesis that supplementing magnesium, in a population at risk for phosphate-induced pathology, in particular on the cardiovascular system, improves outcomes are needed and if this is the consequence of mitigating phosphate toxicity [66].

## 5. The Role of Calcium

As nearly all crystals and mineralized tissues are composed of calcium, besides phosphate, it seems obvious that calcium also modifies phosphate toxicity, at least when ectopic mineralization and calcification is involved. Already at the earliest stages of vascular calcification, calcium acts in synergy with phosphate by inducing the phenotypic switch of contractile VSMC to secretory osteoblast-like cells, in addition to promoting microvesicle release and VSMC apoptosis, both forming a primary nidus for tissue mineralization [67,68]. Moreover, calcium also is a key component of CPPs, and higher calcium concentration increases the formation rate of secondary CPP [21]. These CPP’s may, as outlined above, contribute to VC and may also be involved in destabilizing atherosclerotic plaques, a pathological condition of the endothelial layer [69]. On other aspects also, calcium acts in concert with phosphate. It was shown that phosphate-induced increments of FGF23 are dependent on calcium, and in an experiment in Wistar rats, FGF23 did not increase at all in a setting of hypocalcemia upon exposure to phosphate [28]. Also, in clinical experiments among healthy volunteers, the combined exposure of calcium and phosphate increased FGF23 the most [30]. In a small prospective trial in patients with early stage CKD, adding calcium carbonate to calcitriol treatment, despite being a phosphate binder, increased FGF23 [70], while in other prospective trials, calcium containing binders even aggravated vascular calcification [71,72]. Since FGF23 has been suggested to contribute to complications of CKD, it appears likely that higher calcium and hyperphosphatemia act synergistically on inducing FGF23. In turn, it is conceivable that preventing calcium from rising, phosphate-induced toxicity may be alleviated.

## 6. α-Klotho Mitigates Phosphate Toxicity

When considering the detrimental cardiovascular consequences of phosphate overload, arguably α-klotho qualifies best as an “antidote”. α-Klotho is a protein which is mainly expressed in the kidney, in particular the distal tubule, although there is much debate on the existence of relevant expression at other tissues [73]. It is a membrane-anchored protein with a large ectodomain, which in the kidney, together with fibroblast growth factor receptor 1 (FGFR1) forms the receptor for FGF23 [74] and thereby plays a key role in phosphate homeostasis. Although more early studies reported that only membrane-bound klotho can render FGFR1 as a high affinity receptor for FGF23 [75], a recent report demonstrated that also circulating α-Klotho (after being shed from the plasma membrane) can function as a scaffold, enhancing binding of FGF23 to FGFR1 [76]. Some studies suggested that α-klotho can increase phosphaturia, independent from FGF23, also by modulating the sodium phosphate cotransporter (NaPi2a) expression in the proximal tubule, a key transporter responsible for phosphate reabsorption after its ultrafiltration [77,78,79], but this effect of α-klotho in isolation has been challenged [80]. In CKD α-klotho expression in the kidney is declined, as is its circulating form, soluble α-klotho (sKlotho) [81,82]. The causes of declined expression may be both a loss of renal tissue mass, but also increased hypermethylation of the α-klotho promotor gene, partially silencing it, by uremic toxins [83]. Thereby, in advanced CKD, a vicious circle exists, with decreasing ability to excrete phosphate, which is accompanied by a decline of the protein that provides the best protection against phosphate toxicity. Indeed, several reports show that, besides facilitating phosphate excretion, α-klotho provides cardiovascular protection in the setting of phosphate exposure [84]. This was nicely shown in experiments that demonstrated that arterial wall calcification and even more general features of aging, induced by α-klotho knock-out, could be almost completely rescued by also inducing pronounced phosphaturia, by an additional knock-out of tubular NaPi2a [85,86]. On top of that, this severe aortic calcification in α-Klotho knock-out mice, as a model of both aging and CKD, could be rescued by exogenous α-Klotho delivery by a viral vector [77]. Bringing these intriguing observations one step further, several studies have shed light on the mechanisms of the protection by α-Klotho on vascular calcification. Among the earliest steps in the process of medial layer calcification is a phenotype switch of VSMC to bone forming cells, as discussed above, which is incited, among other stimuli, by hyperphosphatemia. This is accompanied by upregulation of several genes, including *Runx2*, that orchestrate several processes of bone formation. α-Klotho inhibits entrance of phosphate into VSMC through the Pit1-transporter, which is accompanied by suppressed expression of *Runx2,* as shown in Figure 4 [87]. Besides this effect on *Runx2* induced by Pit-1 entrance of phosphate into cells, on a background of α-Klotho deficiency, phosphate also activated AKT/ mammalian target of rapamycin complex 1 (AKT/mTORC1) by phosphate cellular entry, induced vascular calcification and shortened lifespan [88]. Different from the structural abnormalities in the arteries induced by phosphate, this mineral also hampers vasoreactivity by either inducing vasoconstriction directly by its effect on endothelial cells [46,48] or by increased activity of the sympaticoadrenergic axis [89]. These effects too, can be mitigated by α-klotho, since it was shown to be able to increase endothelial cell production of the vasodilating substance nitric oxide [46], and also to promote endothelial cell viability [90].

Besides these effects on arterial vessels or vessel-derived cells, comparable events occur in the aortic valve. Aortic valve calcification in CKD is a clinically very relevant morbidity, that tends to progress more rapidly in these patients than in the general population [91]. In human aortic valve interstitial cells, phosphate induced osteogenic properties of these cells, leading to calcium deposition, was prevented by α-klotho [92]. In addition, the myocardium itself also can be protected by α-klotho from uremia-induced left ventricular hypertrophy and fibrosis [93,94].

Reconciling this plethora of data studying the intricate relation between phosphate and α-klotho, it can be concluded that α-klotho is not only involved in promoting phosphate excretion by the kidney, but also is capable to limit phosphate-induced harm, in particular on the cardiovascular system. The combination of hyperphosphatemia and α-klotho deficiency, as exists in advanced CKD, appears to be a malicious twin. As will be outlined below, focusing on ways to increase α-klotho, if controlling hyperphosphatemia fails, or even more early before phosphate levels rise, might provide novel avenues to an improved outcome in CKD.

## 7. Matrix Gla Protein and Vitamin K Status

Where fetuin A can conceptually be considered as a circulating guard against largely growing calcium-phosphate crystals in the vascular compartment, this function is accomplished at the tissue level by Matrix Gla Protein (MGP) [95]. Like fetuin A, MGP controls and limits crystal growth and can shield small particles, thereby preventing direct exposure of crystals to surrounding tissue. Importantly, this protection against ectopic calcification can only be performed if MGP is carboxylated, a post-translational modification that is fully dependent on vitamin K [96,97]. Therefore, it can be expected that in a setting of vitamin K deficiency, for instance induced by insufficient diets or the use of vitamin K antagonist, phosphate-induced calcification occurs unopposed. Indeed, several observational studies have shown an independent association between the concentration of uncarboxylated MGP, as the functional correlate of vitamin K deficiency, and cardiovascular calcification, both of vessels and valves, and calciphylaxis, an extreme and devastating form of occluding vascular calcification [98,99,100,101,102,103,104]. Based on these findings, clinical trials are ongoing to study the effect of replenishing vitamin K, to improve (phosphate-mediated) ectopic calcification [105,106].

Apart from the specific determination of undercarboxylated MGP, also total MGP has been found to be positively associated with the presence of vascular disease (mainly coronary artery disease or hypertension) [107]. Whether this just reflects a high total ucMGP or a defense attempt [108] requires additional research. Recent evidence reveals a potential role for other proteins than MGP, which also are activated by carboxylation of Gla-moieties on their protein backbone. Especially carboxylated Gla-rich protein (GRP), which appears to have similar protective effects as MGP in protecting form toxicity induced by CPP formation [109].

## 8. Additional Factors that May Modify Phosphate-Toxicity

Besides the above described, and reasonably well-established factors that can either aggravate or relieve pathological changes induced by phosphate, novel effect modifiers emerge. Among these, the trace element zinc is of interest. Zinc was shown, decades ago, to be able to inhibit mineral formation from calcium and phosphate by matrix vesicles [110]. In vitro experiments, using a VSMC resembling cell line, zinc improved cell viability in setting of high P concentration, thereby preventing cell apoptosis, that can form a nidus for initiating matrix mineralization [111]. In three different animal models, prone to calcification, zinc supplementation inhibited vascular calcification, and in that same publication, it was shown that zinc prolonged the formation of secondary CPP (the T50 score) in patients and healthy volunteers with a wide range of eGFR [112]. Noteworthy is that with more severe CKD, zinc levels were lower. A final important observation with regard to the potential role of zinc as phosphate “antidote”, is that its lower intake in the general population was associated with a higher likelihood of the presence of calcification of the abdominal aorta [113].

Besides zinc, the role of either systemic or local acidity may be more important than appreciated [7]. As the local pH dictates the valence of phosphate species (either monovalent H_2_PO_4_^−^, or bivalent HPO_4_^2−^) and phosphate transporters like Pit1 are specific for its valence, the pH also dictates the amount of phosphate entrance into VSMC and other cells [114,115]. The role of acidity, however, is far from settled with some experimental studies reporting potential benefits of lower Ph [116,117,118], while clinical cohort analysis in patients on hemodialysis demonstrated an independent, inverse association between predialysis bicarbonate concentration and coronary artery calcification score [119].

Besides all these factors, that may be targeted clinically, also non-modifiable factors involved in phosphate toxicity have been recognized. These factors may be helpful in identifying those at highest risk and to study what underlies the differences in severity of phosphate-associated morbidity. Among these is gender, where the mortality risk of severe hyperphosphatemia in patients on dialysis is lower for female than male patients [120]. Also, in patients with severe functional impairment, the association between phosphate and risk is less clear [121], possibly because it is overwhelmed by non-phosphate associated complications.

## 9. Clinical Implications for Alleviating Phosphate-Induced Complications

Based on the above, it may be clear that severity of phosphate toxicity is not only dependent on its exposure or concentration, but may in fact be accomplished by mediators, like the formation of CPPs and induction of FGF23, or be highly influences by modifiers, like magnesium, calcium, α-klotho and the vitamin K status. It follows logically that the current scope on lowering the risk associated with hyperphosphatemia may be too constraint. As mediators of disease seldom are the sole consequence of a single exposure, it follows that other ways to lessen the burden that follows the evolution of this mediator may be worthwhile. As mentioned, many other factors than phosphate concentrations are dictating the formation of CPP2. Calcium supplements may aggravate the T50 score [122], while magnesium supplementation and [63,64] and convective dialysis techniques may improve it, compared to techniques solely depending on diffusion [123]. Since fetuin-A and albumin both are able to prolong the T50 score and both are negative acute phase reactants [14], ways to mitigate inflammation may be beneficial, although in clinical practice that may be easier said than done. Likewise, the list of inducers of FGF23 apart from phosphate [124], another potential mediator of phosphate-associated toxicity, is long and includes calcium [70,125], iron deficiency [126,127], inflammation, hyperparathyroidism [40], and high doses of vitamin D. Therefore, careful use of vitamin D and calcium supplements, adequate treatment of over hyperparathyroidism and restoring iron deficiency, may all be of relevance in preventing additional increments of FGF23.

In addition, taking advantage of current knowledge with regard to factors that modify phosphate toxicity might turn out to be beneficial. Magnesium supplements have been shown to alleviate phosphate induced progression of CKD in animal models [128], improve markers of vascular function in trials in human volunteers [56], and slowed progression of coronary artery calcification in patients with predialysis CKD [55]. Importantly, currently widely used magnesium concentration in the dialysate of 0.5mM, aggravates magnesium deficits for most on dialysis, and may need to be increased [129], an approach that is currently being studied [130].

Restoration of α-klotho concentrations is now considered to be a holy grail in CKD research. As hypermethylation of the α-klotho promotor gene is the key mode of silencing of this gene, approaches that demethylate this section on DNA, should be a useful principle and have been shown to increase gene transcription and protein abundance [131,132]. Some studies have shown that under physiological conditions α-klotho is suppressed by the same mechanism of hypermethylation in other tissue like skeletal muscle, which can be reversed, leading to increased expression of α-klotho too, subsequently promoting tissue repair [133]. Analogues of active vitamin D have been shown to be able to increase α-klotho levels, which was associated with a reduction of vascular calcification in an animal model, but the source of α-klotho was not the kidney itself and remained undetermined [134]. This latter finding is in line with the concept that under some ill-defined circumstances other tissues are capable of producing α-klotho as well. That active vitamin D may increase α-klotho by demethylation was suggested by a recent study among kidney transplant recipients [135]. In peripheral mononuclear cells α-klotho expression increased, which was accompanied, and probably preceded by an increase of ration of unmethylated/methylated α-klotho gene.

Dietary supplementation of magnesium, zinc and vitamin K may in the future proof to be simple and cheap interventions, mitigating phosphate toxicity. With regard to vitamin K, a wide range of studies all reported increased carboxylation of MGP by vitamin K, thereby allowing its anti-calcifying to act [100,136,137]. A recent small study demonstrated vitamin K can halt the progression aortic valve calcification [138], and several studies are currently being conducted [106,139].

## 10. Conclusions

The recognition that phosphate-associated toxicity, especially seen in CKD, is at least partially accomplished by intermediate steps, like the formation of CPPs, in particular CPP2, and FGF23, provides the opportunity to address these intermediates instead of, or besides, phosphate only. Moreover, many effect modifiers are operational that influence the pathological events that may follow exposure to phosphate. Modulating these modifiers may be useful, besides directly targeting phosphate, and could include increased intake of vitamin K, magnesium and zinc. The potentially most important defender against phosphate inflicted harm, may be α-klotho. Methods to increase endogenous α-klotho production, even in CKD, are emerging now, and hold promise to change the landscape of treatment of CKD, besides the other listed approaches.

## Figures and Tables

**Figure 1 toxins-11-00522-f001:**
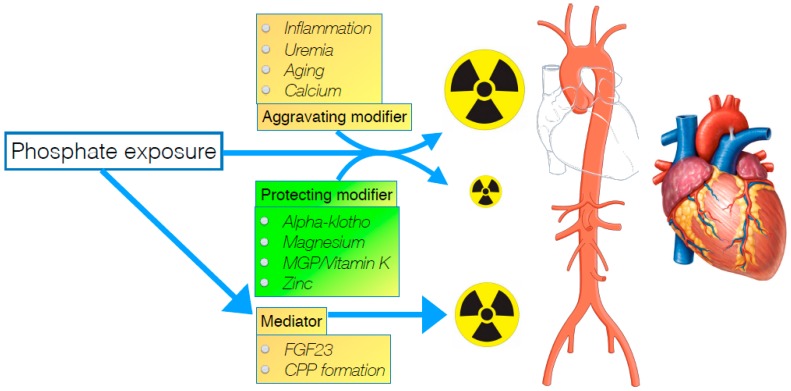
Representation of effect modifiers and mediator of phosphate toxicity on the cardiovascular system at the right. Modifiers can either increase or mitigate direct effects of phosphate on target tissues. Mediators are the consequence of phosphate exposure and inflict harm directly, once formed. CPP: Calciprotein particle. MGP: Matrix Gla Protein.

**Figure 2 toxins-11-00522-f002:**
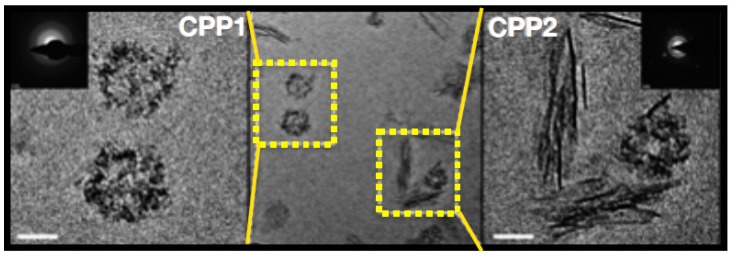
Calciprotein particles (CPP1, left) are spherical particles containing calcium and phosphate ions organized by fetuin-A are covered by fetuin-A, shielding the crystals from the surroundings. Secondary CPP (CPP2) are needle shaped configurations as a consequence of non-organized expanding crystallization as the consequence of either high concentrations of calcium and phosphate, relative deficiency of fetuin A, or otyher metabolic conditions that promote CPP formation, as detailed in the text. Reproduced with permission from Holt et al. 2016, Oxford University Press [20].

**Figure 3 toxins-11-00522-f003:**
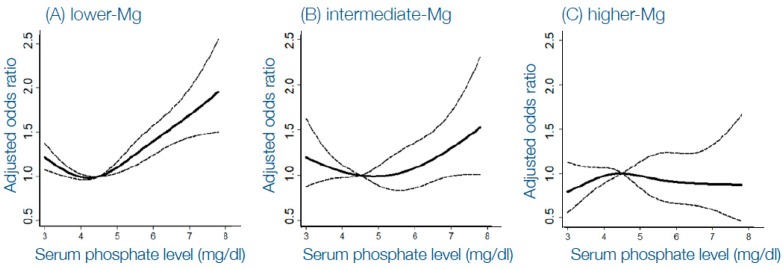
Odds ratio for cardiovascular mortality in a large Japanese cohort of patients on hemodialysis. (**A**) Reflects lower magnesium group, (**B**) intermediate magnesium group and (**C**) higher magnesium group. The dashed line represents the 95% confidence interval. The reference serum phosphate value is 4.5 mg/dL. Reproduced with permission from Sakaguchi et al. [53], 2014, PLoS One.

**Figure 4 toxins-11-00522-f004:**
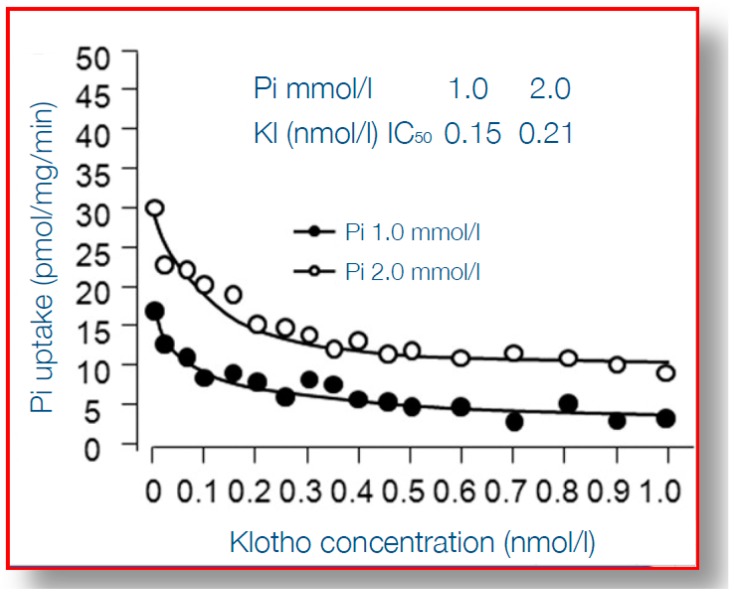
Uptake by vascular smooth muscle cells under varying concentration of α-klotho, and at two different concentrations of inorganic phosphate. On the Y-axis phosphate uptake is shown, on the X-axis concentrations of α-klotho. At higher concentrations α-klotho the uptake is inhibited, for both normal and high phosphate concentration in the medium. Reproduced with permission from Hu et al. [87] 2011, Am Soc Nephrol.

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
