# Peer review of "Modifying Phosphate Toxicity in Chronic Kidney Disease"

_toxins, 2019, doi:10.3390/toxins11090522_

Round 1

Reviewer 1 Report

Comments and Suggestions for Authors

The manuscript is a narrative review that circumscribes through an updated approach the toxicity of phosphate in chronic kidney disease, with focus on factors which can modify the effects of this toxic mineral.

The topic is of great interest for clinicians and basic researchers, and the manuscript is well written in terms of the English language. The paper provides a generous introduction and sufficient background to understand its message. Moreover, the chapters related to the modifiers and mediators of phosphate toxicity are fluent, properly documented and with up-to-date references. 

There are only a few minor issues that should be addressed:

Abstract: There is no specification of the review type (narrative/systematic). Try modifying ”In this overview, several factors are discussed…..” to “In this narrative review several factors are discussed”. Line 28: the term “rachitis” is obsolete. Please change it to “rickets”. There is an abbreviation used without being defined (line 125- LVH). Line 252: the metaplasia of VSMC is not triggered only by hyperphosphatemia. Please insert “among others, by hyperphosphatemia”, or other similar terms. In chapter 7, please refer briefly to parameters which were not approached: total MGP is able to identify patients with vascular diseases (Silaghi et al. Clin Chim Acta. 2019;490:128-134) and Gla rich protein is a promising marker being a constitutive component of the calciprotein particles (Viegas et al. Arterioscler Thromb Vasc Biol. 2018;38(3):575-587). Lines 333-334: the sentence “As mentioned….” is meaningless. Probably “that” was confused with “than”.

It was a true pleasure to read this review from the perspective that the authors did not choose a classical approach on factors that would modify the phosphate toxicity: phosphate binders, calcimimetics, vitamin D sterols and parathyroidectomy.

Author Response

The manuscript is a narrative review that circumscribes through an updated approach the toxicity of phosphate in chronic kidney disease, with focus on factors which can modify the effects of this toxic mineral.

The topic is of great interest for clinicians and basic researchers, and the manuscript is well written in terms of the English language. The paper provides a generous introduction and sufficient background to understand its message. Moreover, the chapters related to the modifiers and mediators of phosphate toxicity are fluent, properly documented and with up-to-date references. 

Author reply:

Many thanks for these remarks

There are only a few minor issues that should be addressed:

Abstract: There is no specification of the review type (narrative/systematic). Try modifying ”In this overview, several factors are discussed…..” to “In this narrative review several factors are discussed”.

Author reply: The reviewer is correct this should be clarified upfront. I now mention in both the abstract and the introduction this is a narrative review

Line 28: the term “rachitis” is obsolete. Please change it to “rickets”.

Author reply: Thanks for pointing to this mistake, which has been corrected in the new version

There is an abbreviation used without being defined (line 125- LVH).

Author reply: This is now written in full on first use of the abbreviation.

Line 252: the metaplasia of VSMC is not triggered only by hyperphosphatemia. Please insert “among others, by hyperphosphatemia”, or other similar terms.

Author reply: Absolutely true. I followed your suggestion

In chapter 7, please refer briefly to parameters which were not approached: total MGP is able to identify patients with vascular diseases (Silaghi et al. Clin Chim Acta. 2019;490:128-134)and Gla rich protein is a promising marker being a constitutive component of the calciprotein particles (Viegas et al. Arterioscler Thromb Vasc Biol. 2018;38(3):575-587).

Author reply: I agree, that there is more to say on the modifying roles of other proteins than ucMGP, that do have comparable modes of action. In the revision the following was added in chapter 7, using the suggested refs:

Apart from the specific determination of undercarboxylated MGP, also total MGP has been found to be positively associated with the presence of vascular disease (mainly coronary artery disease or hypertension).(107). Wether this just reflects a high total ucMGP of a defense attempt(108)requires additional research. Recent evidence reveal a potential role for other proteins than MGP, which also are activated by carboxylation of Gla-moieties on their protein backbone. Especially carboxylated Gla-rich protein (GRP) appears to have similar protective effects as MGP in protecting form toxicity induced by CPP formation(109)

Lines 333-334: the sentence “As mentioned….” is meaningless. Probably “that” was confused with “than”.

Author reply: This indeed is a confusing typo, which now has been corrected. Thanks for spotting this

It was a true pleasure to read this review from the perspective that the authors did not choose a classical approach on factors that would modify the phosphate toxicity: phosphate binders, calcimimetics, vitamin D sterols and parathyroidectomy.

Thank you! That indeed was the goal.

Reviewer 2 Report

My criticism is, that perhaps some other issues were less than optimally considered by the Authors when reviewing toxicity of phosphate.

1.       Bioavailability of dietary phosphorus is highly variable. In particular, modern food processing creating large amount of inorganic phosphorus, highly bioavailable and easily absorbed. Reviewing this issue is perhaps relevant to this paper

2.       Absorption of phosphorus is highly contingent upon availability of activated vitamin-D (calcitriol) or it’s analogues. Supraphysiologic, or sometimes highly supraphysiologic doses of calcitriol (or analogs; extensively used in Nephrology practice of ESRD pateints) are increasing efficacy of Phosphorus absorption immensely, making our P-binder semi-ineffective. It is critical not to confuse the “physiologic” dose of calcitriol supplementation (e.g., 0.25 mcg x3-5 days a week) with pharmacologic dose administration

3.       Somewhat on a related note – this paper neglects to review an important intervention to prevent phosphorus accumulation – the removal of excess parathyroid glands in cases of severe secondary hyperparathyroidism. It is well understood, that after successful parathyroidectomy (PTX) long-term phosphorus is much easier achieved and with lessened pill burden. The very recent concept of “targeted” or “customized” PTX should be particularly reviewed, along with intraoperative PTH monitoring

4.       The major interference of proton-pump inhibitors (PPI) should be in particular recited, as they may cause both hypomagnesemia and ineffectiveness of CaCO3 as both Ca-supplement and P-binder (via achlorhydria). Moreover, CaCO3 is a “de facto” alkaline delivery – but such effect is contingent upon the CaCo3 to CaCl2 conversion, after exposing CaCo3 to HCl acid in the stomach – hence can much be reduced with regular use of PPI.

Author Response

My criticism is, that perhaps some other issues were less than optimally considered by the Authors when reviewing toxicity of phosphate.

1.       Bioavailability of dietary phosphorus is highly variable. In particular, modern food processing creating large amount of inorganic phosphorus, highly bioavailable and easily absorbed. Reviewing this issue is perhaps relevant to this paper

2.       Absorption of phosphorus is highly contingent upon availability of activated vitamin-D (calcitriol) or it’s analogues. Supraphysiologic, or sometimes highly supraphysiologic doses of calcitriol (or analogs; extensively used in Nephrology practice of ESRD pateints) are increasing efficacy of Phosphorus absorption immensely, making our P-binder semi-ineffective. It is critical not to confuse the “physiologic” dose of calcitriol supplementation (e.g., 0.25 mcg x3-5 days a week) with pharmacologic dose administration

3.       Somewhat on a related note – this paper neglects to review an important intervention to prevent phosphorus accumulation – the removal of excess parathyroid glands in cases of severe secondary hyperparathyroidism. It is well understood, that after successful parathyroidectomy (PTX) long-term phosphorus is much easier achieved and with lessened pill burden. The very recent concept of “targeted” or “customized” PTX should be particularly reviewed, along with intraoperative PTH monitoring

4.       The major interference of proton-pump inhibitors (PPI) should be in particular recited, as they may cause both hypomagnesemia and ineffectiveness of CaCO3 as both Ca-supplement and P-binder (via achlorhydria). Moreover, CaCO3 is a “de facto” alkaline delivery – but such effect is contingent upon the CaCo3 to CaCl2 conversion, after exposing CaCo3 to HCl acid in the stomach – hence can much be reduced with regular use of PPI.

Author reply: Based on the valid comments of the reviewer, with all of which i agree, to became clear, that the scope of this narrative review is in need of a bit more clarification. As reviewer 1 pointed rightfully to, the perspective is not on ways to modify phosphate intake or exposure ( the four points of reviewer 2, which I recently reviewed for Kidney Int), but on co-existing factors that modify or mediate severity of phosphate toxicity for a given concentration or intake. To clarify this I added the following text to the introduction, which the reviewer might consider sufficient now:

"Modes to control hyperphosphatemia or phosphate intake, including dietary intervention, phosphate binders, phosphate transport inhibitors, controlling hyperparathyroidism, restricted use of (high dosed) active vitamin D, and adapting dialysis schemes if applicable, have been reviewed recently and are beyond the scope of this review(7)".

Round 2

Reviewer 2 Report

The paper is now acceptable in the present form.